# ARTIPG++: TOWARDS EFFICIENT PROCEDURAL GENERATION OF ARTICULATED OBJECTS AND ANNOTATIONS

## ABSTRACT

To leverage deep learning in advancing vision perception and embodied intelligence, an extensive number of high-quality and richly annotated 3D articulated objects is essential. However, current methods for collecting articulated objects and their annotations are either based on human effort or physics simulators, which are difficult to scale up, posing challenges to the collection of large-scale and richly annotated articulated objects. In such context, procedural generation has recently gained attention in articulated object synthesis. However, it still faces challenges such as reliance on external assets and the complexity of designing procedural rules. To this end, we propose ArtiPG++, a highly efficient framework for synthesizing articulated objects with rich annotations, featuring three key advantages: 1) asset-free spatial structure synthesis via procedural rules, 2) labor-free synthesis of realistic geometric details, along with precise and diverse annotations, and 3) easy expansion to new object categories, with a ready-to-use tool for convenient synthesis. ArtiPG++ currently supports the procedural synthesis for 39 common object categories, and requires only a few hours to develop procedural generation rules for novel categories, which is a one-time effort for infinite objects synthesis. We conduct extensive evaluations on the objects and annotations synthesized by ArtiPG++, through both direct comparisons in terms of diversity and distribution, as well as performance in downstream tasks. **Please refer to the appendix for more details, analysis, discussions and code implementation.**

## 1 INTRODUCTION

High-quality 3D articulated object data has become increasingly important in AI research, particularly in the fields of 3D computer vision (Zhao et al., 2021; Qi et al., 2017; Xiang et al., 2021; Wang et al., 2019; Geng et al., 2023) and robotic manipulation (Mo et al., 2021; Ning et al., 2023; Mousavian et al., 2019; Fang et al., 2023) in today's data-driven era. However, collecting real-world articulated object data and annotating it for various tasks remains costly and time-consuming (Mo et al., 2019; Liu et al., 2022), posing a significant challenge to scaling training datasets for articulated object learning. Meanwhile, current articulated object synthesis methods either rely on external assets as references during generation (Sun et al., 2024; Liu et al., 2024), or fail to automatically and precisely produce annotations for diverse tasks (Lei et al., 2023; Liu et al., 2024; Gao et al., 2025).

To address these limitations, we propose ArtiPG++, a fully procedural pipeline for synthesizing large number of realistic articulated objects with rich annotations. Compared to existing procedural generation approaches, ArtiPG++ offers three key advantages: 1) **Asset-free Procedure:** ArtiPG++ synthesizes articulated object structures entirely from scratch, without relying on external assets during synthesis. 2) **Realistic and Diverse Synthesis with Annotations:** Unlimited variation in spatial structures is achieved through procedural design, while a generative model enhances realism and diversity of geometric details. Mathematically defined rules further enable precise and diverse annotation synthesis. 3) **Scalability and Ease of Use:** ArtiPG++ supports 39 object categories, providing fine-grained diversity. Designing a new part template typically takes about one hour, while scripting a new object category requires around eight hours, making the system highly scalable and generalizable. We have also integrated ArtiPG++ into a ready-to-use tool, allowing researchers to easily synthesize articulated objects with a single command.

Thanks to its procedural foundation, ArtiPG++ can analytically define various types of knowledge, allowing for the synthesis of annotations across a wide range of tasks on articulated objects. We have also developed a user-friendly interface that enables users to generate large sets of objects with a single command. ArtiPG++ supports: **1)** synthesizing point clouds with annotated point-wise labels for downstream tasks, **2)** synthesizing URDF-format objects, enabling integration with physics simulators for data collection or manipulation tasks, and **3)** retrieving spatial structural parameters for further research. These capabilities make ArtiPG++ a highly versatile tool that significantly reduces the barrier to entry for articulated object learning.

Finally, we evaluate ArtiPG++ across 39 object categories from open-source articulated object datasets (Mo et al., 2019; Xiang et al., 2020; Chang et al., 2015). Using direct evaluation on synthesis quality, along with four downstream tasks (part segmentation, point cloud completion, part pose estimation, and robotic manipulation), we comprehensively demonstrate ArtiPG++'s superiority.

In summary, towards efficiently collecting high-quality data and annotations for articulated objects, our contributions are follows: **1)** We organize a novel framework for procedural generation of object spatial structures and factor the implementation code, producing objects with more diverse structural topologies, improving runtime efficiency, and enhancing scalability and maintainability. **2)** We propose to learn a distribution of geometric details conditioned on an object's spatial structure via a cross-category generative model, enabling the synthesis of realistic and diverse details that align precisely with the structure, and support accurate grounding of mathematically defined annotations. **3)** We address Arti-PG (Sun et al., 2024)'s reliance on external assets during synthesis by restructuring procedural rules for spatial structures and employing generative models for geometric detail synthesis, mitigating manual efforts and acclerating synthesis by 10 times. Our method expands supported categories by 1.5 times towards broader application, and synthesizes higher-quality objects, improving manipulation performance on unseen categories by at least 13%.

## 2    RELATED WORKS

### 2.1    ARTICULATED OBJECT SYNTHESIS

Beyond CAD design and real-world scanning, articulated object synthesis has gained attention. A common paradigm employs generative models trained on real data to generate new shapes (Achlioptas et al., 2018; Fan et al., 2017; Vahdat et al., 2022; Groueix et al., 2018; Nash et al., 2020; Lei et al., 2023; Liu et al., 2024; Gao et al., 2025). For example, NAP (Lei et al., 2023) and CAGE (Liu et al., 2024) generate articulated objects from reference geometry graphs, while MeshArt (Gao et al., 2025) uses a decoder-only approach to retrieve shapes. However, these methods are limited by scarce 3D data and lack automatic and diverse annotations. Vision-language models (Radford et al., 2021; Achiam et al., 2023; Liu et al., 2023b; Siddiqui et al., 2024) improve generalization but often yield low-diversity or physically implausible shapes, and still requiring manual labeling. Recently, procedural generation (Raistrick et al., 2024; Sun et al., 2024; Morrison et al., 2020)has gained traction. For example, Infinigen Indoor (Raistrick et al., 2024) produces diverse and realistic objects, but lacks object-level annotations and depends on category-specific rules, limiting scalability.

Arti-PG (Sun et al., 2024) is a recent procedural generation method that synthesizes articulated objects with rich annotations by: 1) modifying spatial structures of manually annotated assets to create new shapes, 2) transferring geometric details from these assets, and 3) analytically defining knowledge annotations from parameterized structures. While Sun et al. (2024) provides an effective foundation for articulated objects' procedural generation, it relies heavily on external assets, limiting scalability and efficiency. In contrast, our method synthesizes spatial structures from scratch, eliminating asset dependency, enhancing flexibility, and improving realism through model-based geometric detail synthesis. ArtiPG++ also expands category coverage and annotations, providing a more diverse, autonomous, and extensible framework for articulated object synthesis.

### 2.2    3D ARTICULATED OBJECT UNDERSTANDING TASKS

Humans interact with a diverse range of 3D articulated objects in daily life. Understanding articulated objects is a fundamental step for intelligent agents in understanding the 3D world. To facilitate this understanding, a series of vision and robotic tasks have been studied.

**Vision Tasks.** Part segmentation (Qi et al., 2017; Zhao et al., 2021) assigns labels to point clouds to distinguish different object parts. Part pose estimation (Geng et al., 2023; Liu et al., 2023a) predicts the 6-dof transformations of parts in 3D space, including translation, rotation, and scale. Both tasks require a strong understanding of the object's spatial structure. Point cloud completion (Tchapmi et al., 2019; Wen et al., 2020; Xiang et al., 2021; Yuan et al., 2018; Groueix et al., 2018) focuses on reconstructing an object's full shape from partial observations, emphasizing understanding on geometric details besides the spatial structures.

**Manipulation Tasks.** 3D object manipulation (Mo et al., 2021; Ning et al., 2023; Geng et al., 2023) involves tasks where embodied agents interact with objects according to specific instructions. Where2Act (Mo et al., 2021) predicts per-pixel action likelihoods and generates manipulation proposals. Where2Explore (Ning et al., 2023) introduces a few-shot learning framework that transfers manipulation knowledge to novel objects by measuring affordance similarity across categories. GAPartNet (Geng et al., 2023) provides a dataset annotated with semantic and affordance labels and proposes manipulation based on actionable parts. ManipLLM (Li et al., 2024) uses LLM's common-sense reasoning capabilities to infer objects' affordances. These strategies rely heavily on a solid understanding of object pose and affordances.

In this paper, we conduct extensive experiments across these tasks to comprehensively evaluate the quality of our synthetic training data regarding spatial structure, geometric detail, and annotations.

## 2.3 KNOWLEDGE ACQUISITION ON 3D OBJECTS

Various types of knowledge can be defined on 3D objects such as semantics, poses, affordances, surface friction, and textures (Mo et al., 2019; Geng et al., 2023; Liu et al., 2022; Chang et al., 2015; Xiang et al., 2020). Understanding this knowledge is essential for intelligent agents to interact effectively with 3D objects. However, acquiring such knowledge still heavily depends on labor-intensive manual annotations, particularly for point-wise labels like semantics, part segmentation, and affordances, limiting scalability. Traditional annotation methods require annotators to label 3D shapes on 2D screens by frequently adjusting the viewpoint (Geiger et al., 2013). PartNet (Mo et al., 2019) introduced a web-based tool where annotators answer guided questions and manually mark part geometries. ShapeNet (Chang et al., 2015) used a hybrid approach combining algorithmic predictions with human verification. GAPartNet (Geng et al., 2023) relied entirely on human annotations and applied extensive heuristics to clean data, define semantics, and fit part poses with oriented bounding boxes.

In addition to manual annotation, some automatic approaches using physics simulators have been explored (Pinto & Gupta, 2016; Lohmann et al., 2020; Khansari et al., 2020; Murali et al., 2020; Mo et al., 2021; Ning et al., 2023; Pathak et al., 2018). For example, Pathak et al. (2018) acquires segmentation knowledge through physical interaction, but the complexity of actions limits its scalability. Where2Act (Mo et al., 2021) learns affordances by performing random interactions with 3D objects in the SAPIEN simulator (Xiang et al., 2020), but suffers from low efficiency in certain settings that only 1% of samples are positive for pulling actions. Thus, a simple, efficient, and scalable method for automatically generating 3D object annotations remains an open research challenge.

## 3 METHODS

### 3.1 OVERVIEW

We build on Arti-PG (Sun et al., 2024) to represent articulated objects through their spatial structures and geometric details. The spatial structure is defined as a composition of geometric shapes, each parameterized by templates, with connections encoded via binary descriptors. These templates provide analytical definitions of structural knowledge, while geometric details are modeled as offsets applied to point clouds sampled from the structure, producing realistic object representations.

To overcome the limitations of existing procedural generators, we design new procedural rules that synthesize spatial structures entirely from scratch. This removes reliance on manually annotated data and allows the generation of diverse object categories. In addition, a diffusion model trained on real objects is employed to generate varied and realistic geometric details. These process enable

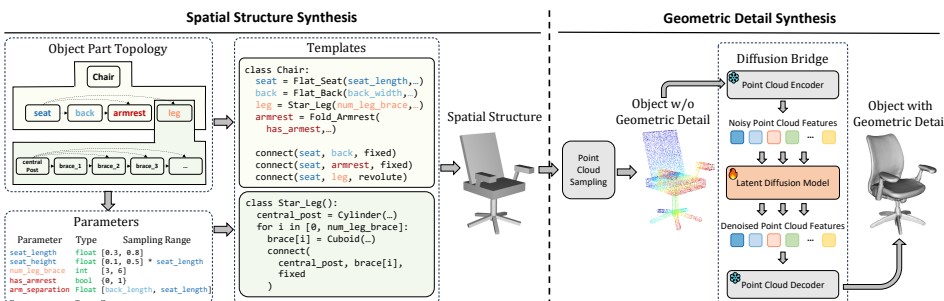

Figure 1: Overview of the ArtiPG++ pipeline. [Left] Spatial structure synthesis comprises parameters and templates, which are synthesized and instantiated in the topological order of object parts. [Right] Geometric details are synthesized via a diffusion bridge. The input point clouds sampled from the synthesized spatial structure are encoded into a latent space, enhanced with the denoising module, and then decoded back to point clouds with geometric details.

us to integrate spatial structures and geometric details into rendered meshes and apply standard methods to synthesize textures without relying on external assets. **Please refer to the appendix for more details, discussions and implementation of our method.**

## 3.2 SPATIAL STRUCTURE SYNTHESIS

In this section, we introduce the procedural generation rules for synthesizing object spatial structures without relying on external assets. These rules enable the synthesis of diverse objects within a given category. As illustrated in Fig. 1, the process starts from an object category (*e.g.*, drink bottle, thermos, coffee mug) with its part topology graph loaded, represented as a directed acyclic graph (DAG) where nodes denote parts and edges define their connections. We then traverse the parts in topological order where for each part, a template is selected from its candidate template set, and the template's parameters are instantiated either by sampling within predefined ranges or by constraining the values based on parameters of predecessor parts. Each instantiated template defines the geometry of its part, and all parts are finally combined according to the DAG connections to yield the complete spatial structure of the object. The psuedo code for synthesizing objects' spatial sturcture is provided below. Detailed codes and explanations featuring spatial structure synthesis of a bucket is provided in the appendix.

---

**Algorithm 1:** Pseudo code for Procedurally Synthesizing Spatial Structure

---

**Require:** Object Category $C$
**Ensure:** Object Spatial Structure $O$
  1: Load the part topology graph $G = (V, E)$ corresponding to category $C$, where $G$ is a DAG
  2: Traverse each part $v \in V$ in the topological order of $G$
  3:    Select a template $t$ from the template set $T_v$ of part $v$
  4:    For each parameter $p$ in template $t$
  5:       Determine the valid range of $p$ based on predecessor parts or predefined ranges
  6:       Randomly sample a value within the range and assign it to $p$
  7:    Instantiate template $t$ with the chosen parameters to create the geometry of part $v$
  8: Combine all instantiated parts into the final object $O$
  9: **return** $O$

---

To ensure physical plausibility, we perform a breadth-first traversal of the synthesized object to detect local errors (*e.g.*, part penetration or floating). Simulated annealing then refines parameters of problematic templates through small perturbations until a valid structure is achieved.

The synthesized spatial structure is rendered as a unified mesh. Additionally, it can be serialized into a URDF format by rendering individual part meshes and assembling them according to their binary descriptors. This enables seamless integration with physics simulators for physical validation, manipulation tasks, or interaction data collection.

### 3.3 GEOMETRIC DETAIL SYNTHESIS

We leverage a diffusion bridge (Li et al., 2025) to transform the synthesized spatial structure into a mesh with geometric details. We train the diffusion bridge using paired data consisting of: i) real-world object meshes with geometric details and ii) their corresponding spatial structures. This enables the model to learn geometric details from real reference objects and synthesize geometric details conditioned on the spatial structure.

Specifically, we first sample points from the paired meshes to generate point clouds, and then adopt a point cloud encoder (we use the DoRA VAE (Chen et al., 2025)) to encode the point cloud into a latent space. The latent representation is denoted as $\mathbf{Z} \in \mathbb{R}^{N \times D}$, where $N$ is the ds number of sampled points and $D$ is the latent feature dimension. Subsequently, we use the latent representation of the point cloud of spatial structure as both the initial state and the condition for the diffusion bridge, and train the model to reconstruct the latent representation of the point cloud sampled from the object with geometric details as the target.

For the diffusion model, we adopt a DiT-based denoising network, whose architecture follows the design of Craftsman (Li et al., 2025). During training, the parameters of the DoRA VAE are kept frozen, and only the DiT denoiser is optimized. Since the latent produced by DoRA is permutation-invariant, the ordering of the $N$ points within $\mathbf{Z}$ is not consistent across encodings, which makes it non-trivial to directly measure the reconstruction error between the latent of the spatial structure and that of the detailed object. To resolve this correspondence issue, we employ Earth Mover's Distance (EMD) matching to align the latent representations from the spatial structure and the detailed object along the point dimension $N$, ensuring consistent supervision during training.

To synthesize geometric details, we input the latent representation of point clouds from the spatial structure into the diffusion bridge. The output is then decoded into an unsigned distance field using (Chen et al., 2025), which is further converted into an object with geometric details.

### 3.4 ANALYTIC KNOWLEDGE ALIGNMENT

We construct the spatial structure of objects using advanced templates as fundamental components. Each point in the synthesized object is analytically associated with a corresponding advanced template. Leveraging this property, we can mathematically define knowledge on these templates, including pose-based and region-based knowledge, and analytically align region-based knowledge to the object's point cloud.

**Region-based Knowledge.** It consists of knowledge (*e.g.*, semantic, instance, affordance) defined on an object's point cloud. Region-based knowledge is first expressed as mathematical functions over template parameters, as shown in Fig. 2-[Left]. For each type, we locate points within the region and its neighborhood on the detailed point cloud and assign corresponding labels. This process propagates region-based knowledge from structure to point cloud, enabling synthesis of diverse annotations without extra human effort.

**Pose-based Knowledge.** As shown in Fig. 2-[Right], pose-based knowledge (*e.g.* part pose, grasp pose, *etc.*) can be analytically expressed as an $SE(3)$ transformation matrix. The pose is initially defined in a geometry template's local coordinates, and then mapped to the advanced template's coordinate by applying its translation and rotation, and finally transformed into world coordinate using object's global pose. Such formulation enables automatic annotation on diverse object instances that share same templates.

### 3.5 EFFICIENT DESIGN OF ARTIPG++

Following Arti-PG (Sun et al., 2024), we define 10 reusable geometry templates. The hierarchical design of our advanced templates enables them to be flexibly constructed by combining these predefined geometric components. Practically, this construction is efficiently implemented through code inheritance and invocation, allowing for rapid and systematic template creation. On average, designing a new advanced template takes approximately 2 hours.

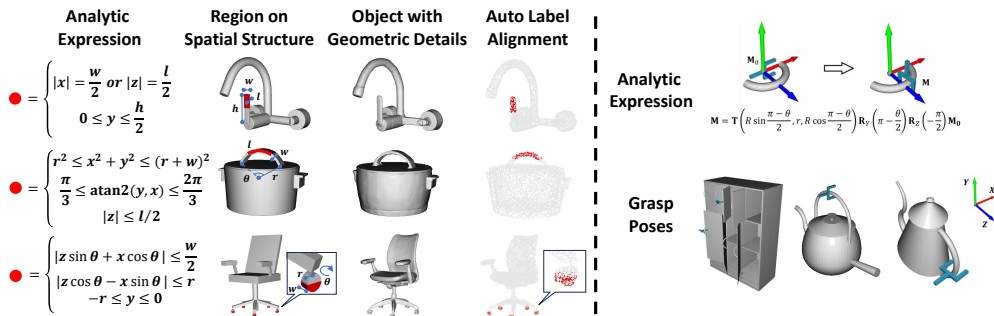

Figure 2: **[Left]** Demonstration of region-based knowledge alignment on a faucet, pot and chair, through analytically expressed regions using template parameters. **[Right]** Demonstration of grippers' grasp pose on torus handle, showcasing pose-based knowledge alignment implemented from local coordinates to world coordinates.

Moreover, defining advanced templates, procedural rules for spatial structure synthesis, and mathematical formulations for object knowledge (*e.g.* semantics, instances, and affordances) requires only basic programming and elementary mathematical skills. To evaluate the accessibility and usability of our pipeline, we recruited several first-year college students to independently design procedural generation rules and knowledge descriptions for 6 different object categories. A separate group of volunteers with similar experience implemented the same tasks using Arti-PG for comparison. Using ArtiPG++, the average time to complete the full pipeline was only 8.7 hours, which is a one-time effort that enables infinite synthesis of high-quality objects and annotations. Furthermore, our pipeline generates a single object in just 0.015 seconds on average with Intel(R) Xeon(R) Gold 6133 CPU and single 4090 GPU, showcasing its exceptional efficiency and scalability for large-scale articulated object and annotation creation.

## 4 EXPERIMENTS

We evaluate the effectiveness of our approach through extensive experiments. First, we compare our method with classic articulated object synthesis approaches (Lei et al., 2023; Liu et al., 2024) and the state-of-the-art one (Gao et al., 2025), to demonstrate its superior synthesis quality. Next, we assess the quality of the synthesized objects and annotations by comparing the performance of neural networks trained on synthesized data versus real data across various visual and manipulation tasks. The visual tasks include part segmentation, point cloud completion, and part pose estimation. The manipulation tasks involve guiding embodied agents to interact appropriately with objects. In the following sections, we introduce each task, the main results and analysis. **Please refer to the appendix for detailed task settings, experimental results, and additional discussions.**

### 4.1 ARTICULATION SYNTHESIS QUALITY

Following Lei et al. (2023); Liu et al. (2024); Gao et al. (2025), we evaluate synthesis quality using Instantiation Distance (ID), defined with Chamfer-L1 distance on 2,048 sampled surface points. From ID, we compute Minimum Matching Distance (MMD), Coverage (COV), and 1-Nearest-Neighbor Accuracy (1-NNA). We include classic methods NAP (Lei et al., 2023), CAGE (Liu et al., 2024), and the state-of-the-art method MeshArt (Gao et al., 2025) as baselines. For fairness, *chair*, *table*, and *storage furniture* from PartNet-Mobiilty (Xiang et al., 2020) are randomly split into training and testing sets, and these learning-based methods are re-trained on the same dataset following these methods respectively. Specifically, our approach requires no training, as objects are procedurally synthesized, with procedural rules designed using PartNet-Mobility Xiang et al. (2020) objects as reference. Since FID cannot be directly computed from 3D representations, we follow Qi et al. (2017) to train a classifier on the 3 object categories and use its encoder to extract feature vectors from objects. These encoded features are then used to calculate FID scores. As shown in Tab. 1, our method consistently outperforms the baselines across all three categories on COV, MMD, and 1-NNA, while achieving a comparable FID to the best baseline.

Table 1: Experimentatl results of articulation synthesis quality. Lower MMD indicates better quality, higher COV indicates broader coverage of reference objects. Specifically, 50% 1-NNA is optimal, reflecting closer distribution compared with reference objects. COV, MMD, and 1-NNA are in percentage. FID is multiplied by $10^3$.

| Method | Chair | | | | Table | | | | Storage Furniture | | | |
|---|---|---|---|---|---|---|---|---|---|---|---|---|
| | COV↑ | MMD↓ | 1-NNA | FID↓ | COV↑ | MMD↓ | 1-NNA | FID↓ | COV↑ | MMD↓ | 1-NNA | FID↓ |
| NAP | 47.3 | 7.7 | 72.6 | 30.85 | 33.4 | 9.2 | 60.8 | 27.70 | 39.6 | 5.3 | 67.6 | 19.85 |
| CAGE | 46.0 | 5.2 | 72.2 | 10.12 | 36.7 | 6.7 | 63.9 | 9.35 | 44.1 | 3.8 | 65.4 | 7.92 |
| MeshArt | 56.0 | 5.4 | 67.7 | 4.85 | 43.9 | 6.2 | 62.8 | 15.62 | 50.8 | 4.2 | 65.1 | 6.40 |
| Ours | 72.1 | 4.2 | 65.3 | 5.25 | 60.2 | 5.3 | 58.1 | 5.98 | 62.4 | 2.2 | 59.4 | 6.08 |

In addition to evaluating articulation synthesis quality among different methods, we also provide a quantitative analysis of object diversity and distribution using ArtiPG++, compared with real objects (Xiang et al., 2020) across all 39 object categories. Please refer to appendix for the detailed results and analysis.

## 4.2 DOWNSTREAM TASKS: VISION

**Part Segmentation.** We collect 26 categories of real-world objects with well-defined parts from Mo et al. (2019); Xiang et al. (2020) and use their part annotations as ground truth for the part segmentation task. The labels for synthesized objects are obtained as described in Sec. 3.4. We employ the classical and widely used Point-Transformer (Zhao et al., 2021) as the neural network model, and randomly sample 2048 points from the point clouds of both real and synthesized objects as input. The mean accuracy (**mAcc**) and mean IoU (**mIoU**) are adopted as metrics for this task.

**Point Cloud Completion.** We collect 39 categories of real-world objects from Mo et al. (2019); Xiang et al. (2020); Chang et al. (2015); Liu et al. (2022). For both real and synthesized objects, we follow Yuan et al. (2018) to uniformly sample 16,384 points for complete point clouds and generate partial point clouds by projecting them from 8 viewpoints. SnowflakeNet (Xiang et al., 2021) is used as the backbone model, with 2,048 points randomly sampled from partial clouds as input. Following Xiang et al. (2021), Chamfer Distance (CD) is used as the evaluation metric.

**Part Pose Estimation.** We adopt NPCS from GAPartNet (Geng et al., 2023) as the ground truth for real-world objects in part pose estimation, and derive ground truth for synthesized objects by inversely applying translation and rotation to each part based on their parameters. Following Geng et al. (2023), we report evaluation metrics including translation error ($T_e$), rotation error ($R_e$), scale error ($S_e$), 3D mIoU, and the "5°/5cm" accuracy ($A_5$).

**Main Results.** For each task, we use real-world and synthesized objects with corresponding annotations as training data. The synthesized objects are generated by Arti-PG (Sun et al., 2024) and our method. We follow baseline training procedures and keep all conditions identical except for the training data. Tab. 2 presents the main results across the three vision tasks. We observe consistent performance improvements over models trained on both real-world and Arti-PG synthesized data, with notable gains exceeding 9% in **mIoU** and Chamfer Distance (CD). These results demonstrate that our method effectively synthesizes high-quality objects with accurate annotations.

Table 2: Experimental results of vision tasks, including part segmentation, point cloud completion and part pose estimation.

| Metrics | Segmentation | | Completion | Part Pose Estimation | | | | |
|---|---|---|---|---|---|---|---|---|
| | mAcc(%) ↑ | mIoU(%) ↑ | CD($\times10^{-4}$cm)↓ | $R_e$(°) ↓ | $T_e$(cm)↓ | $S_e$(cm)↓ | mIoU(%) ↑ | $A_5$(%) ↑ |
| Real Obj | 89.5 | 74.5 | 11.3 | 11.0 | 0.043 | 0.025 | 44.1 | 24.8 |
| Arti-PG | 91.3 | 79.4 | 10.4 | 10.5 | 0.039 | 0.022 | 48.3 | 25.9 |
| Ours | 91.8 | 81.3 | 9.2 | 10.2 | 0.035 | 0.020 | 49.0 | 26.1 |

## 4.3 Downstream Tasks: Manipulation

In this section, we further evaluate the effectiveness of our method on manipulation tasks, which are closely tied to embodied AI. We select a variety of frameworks (Mo et al., 2021; Ning et al., 2023; Geng et al., 2023) that represent different paradigms for learning manipulation. Specifically, Where2Act (Mo et al., 2021) and Where2Explore (Ning et al., 2023) focus on learning pixel-wise affordance maps, while GAPartNet (Geng et al., 2023) predicts the poses of actionable parts to guide manipulation actions. Additionally, ManipLLM (Li et al., 2024) leverages the reasoning capabilities of large language models to infer affordances. We evaluate manipulation performance using success rate, an objective and stringent metric. Thus, these tasks offer a comprehensive measure of both the quality of synthesized objects and the accuracy of their annotations.

**Main Results.** We collect 972 objects across 15 categories from real-world datasets (Xiang et al., 2020), splitting them into training and testing sets following Arti-PG (Sun et al., 2024). We also synthesize objects from the same categories using both Arti-PG (Sun et al., 2024) and our approach. For Where2Act (Mo et al., 2021) and Where2Explore (Ning et al., 2023), we mathematically define affordance knowledge and assign it to the synthesized objects. For GAPartNet (Geng et al., 2023), we first obtain the translation and rotation parameters of the target parts as described in Sec. 3.4, then apply the inverse transformations to generate the NPCS representations. For ManipLLM (Li et al., 2024), affordances on point clouds are mathematically defined and propagated to pixel-wise affordance on RGB images for training. We compare the performance of manipulation frameworks trained on real-world data with original annotations, data and annotations synthesized by Arti-PG, and data synthesized by our method, as shown in Tab. 3. These results demonstrate the effectiveness of our approach in generating high-quality affordance annotations, including affordance regions and part poses, which effectively drive manipulation models across different paradigms.

Table 3: Experimental results of manipulation tasks on training / testing categories. All values are percentage.

| Training Source | Where2Act | Where2Explore | GAPartNet | ManipLLM |
|---|---|---|---|---|
| PartNet-Mobility | 26.1 / 14.4 | 26.9 / 20.5 | 29.7 / 29.0 | 32.0 / 30.6 |
| Arti-PG | 26.7 / 16.9 | 28.0 / 25.7 | 32.8 / 30.1 | 33.5 / 31.6 |
| ArtiPG++(Ours) | 29.0 / 20.8 | 32.1 / 30.2 | 35.7 / 34.2 | 36.4 / 36.1 |

## 4.4 Ablation Study

**Training Data Collection Method.** In addition to synthesizing affordance knowledge through analytic knowledge alignment, our method supports generating URDF-format objects for simulator-based affordance data collection. To evaluate the effectiveness of these two approaches, we follow the collection and training protocols of Where2Act (Mo et al., 2021) and ManipLLM (Li et al., 2024). Results in Tab. 4 show that analytic alignment yields more precise affordance labels for model training. This advantage is also evident in Fig. 3, where affordance labels derived from analytic alignment show greater consistency, completeness, precision compared to those obtained via simulation. Moreover, data collection through simulation requires over 10 hours, whereas analytic alignment synthesizes affordances in only about 6 seconds per object on average. Together, these findings demonstrate that our method not only improves label quality but also offers a far more efficient pipeline for generating affordance training data.

Table 4: Ablation study on collection method of affordance. The results are reported on training / testing categories. All values are percentage.

| Metrics | Where2Act | ManipLLM |
|---|---|---|
| simulated affordance | 26.8 / 18.5 | 33.4 / 31.0 |
| analytic affordance | 29.0 / 20.8 | 36.4 / 36.1 |

**Module Contribution Analysis.** The spatial structure synthesis module is the core of our object synthesis pipeline. After generating the spatial structure, we apply a physics validation module to

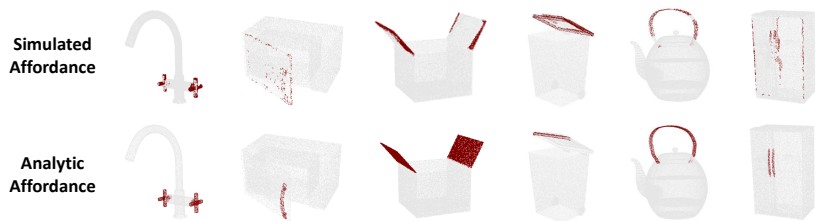

Figure 3: Visualization of affordance collection using simulation and analytic knowledge alignment. Red points indicate affordance regions on the objects.

ensure physical plausibility and a geometric detail synthesis module to enhance realism. To analyze their contributions, we remove each module in turn and synthesize objects. Using the settings from Sec. 4.2 and adopting ManipLLM (Li et al., 2024) for manipulation tasks (Sec. 4.3), we conduct experiments with results shown in Tab. 5. Overall, both modules improve performance across all tasks. Specifically, physics validation has a greater effect on part segmentation and pose estimation, while geometric detail synthesis contributes more to point cloud completion. Their impact is especially pronounced in the more comprehensive manipulation tasks.

Table 5: Ablation study on geometric detail synthesis (GD) and physics validation (PV).

| Metrics | Segmentation | | Completion | Part Pose Estimation | | Manipulation |
|---|---|---|---|---|---|---|
| | **mAcc**(%) ↑ | **mIoU**(%) ↑ | CD($\times 10^{-4}$cm)↓ | **mIoU**(%) ↑ | $A_5$(%) ↑ | ssr(%) ↑ |
| w/o GD | 90.2 | 75.8 | 11.5 | 46.6 | 25.3 | 25.8 / 20.2 |
| w/o PV | 86.5 | 74.5 | 13.2 | 42.8 | 23.5 | 31.2 / 28.9 |
| Full Method | 91.8 | 81.2 | 9.2 | 49.0 | 26.1 | 36.4 / 36.1 |

**Feature Extractor in Geometric Detail Synthesis.** To synthesize geometric details, we use Dora (Chen et al., 2025) to extract features of geometric details. In this section, we analyze the performance of different feature extractors by comparing PointNet++ (Qi et al., 2017), Craftsman VAE (Li et al., 2025), and Dora (Chen et al., 2025). To ensure fair comparison, we train PointNet++ as classifiers on 39 object categories to optimize their feature extraction. The ablation results in Tab. 6 show that the choice of feature extractor significantly affects the quality of synthesized objects, with Dora achieving the best performance in our pipeline. **Please refer to the appendix for visualization of ablation on geometric detail synthesis and their impact on label alignment.**

Table 6: Ablation study on the feature extractor involved in geometric detail synthesis.

| Metrics | Segmentation | | Completion | Part Pose Estimation | | Manipulation |
|---|---|---|---|---|---|---|
| | **mAcc**(%) ↑ | **mIoU**(%) ↑ | CD($\times 10^{-4}$cm)↓ | **mIoU**(%) ↑ | $A_5$(%) ↑ | ssr(%) ↑ |
| PointNet++ | 91.3 | 80.6 | 10.1 | 47.6 | 25.4 | 34.6 / 31.8 |
| Craftsman | 91.6 | 81.2 | 9.4 | 48.2 | 26.0 | 36.0 / 35.5 |
| Dora | 91.8 | 81.2 | 9.2 | 49.0 | 26.1 | 36.4 / 36.1 |

## 5 CONCLUSION

In this paper, we propose ArtiPG++, a procedural framework for synthesizing a large, diverse set of articulated objects, richly annotated for various vision and embodied AI tasks. Our method defines procedural rules to generate spatial structures of articulated objects from scratch, removing reliance on external assets. We enhance these objects by adding detailed geometric features through a learned generative diffusion model. This approach significantly improves the diversity, realism, efficiency, and scalability of articulated object synthesis compared to prior work. Extensive experiments across visual and manipulation tasks demonstrate the effectiveness and superiority of our method. We will expand the range of supported categories to further unlock scalable, detailed understanding of articulated objects.

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

# A APPENDIX

We provide comprehensive appendices to support our main paper and further demonstrate the effectiveness and efficiency of our approach. We have included several points in the appendix which could not be elaborated in detail in the main paper due to space constraints. The appendix is organized as follows:

- Appendix. A.1: Implementation of the procedural synthesis of ArtiPG++.
- Appendix. A.2: Discussions and analysis of efficiency and complexity of synthesized structure of ArtiPG++.
- Appendix. A.3: Detailed discussions, settings, and results of the main experiments and ablation study involved in the main paper.
- Appendix. A.4: Demonstration of real-world experiments.
- Appendix. A.5: Implementation of our approach in Python using "Bucket" as an example.
- Appendix. A.6: Discussion and visualization of the introducing texture on our synthesized articulated objects.
- Appendix. A.7: Qualitative comparison with other articulated object synthesis methods.
- Appendix. A.8: Visualization of synthesized objects.
- Appendix. A.9: Statement of usage on LLMs.

We also provide code examples in the "code-examples" folder in the supplementary material, containing Chair, Kettle, and Microwave. Please check them for directly synthesizing articulated objects using our method.

## A.1 IMPLEMENTATION OF ARTIPG++

**Spatial Structure Synthesis Implementation.** To ensure that our approach synthesizes objects with real-world variety, diversity, and realism, we use common open-source datasets (Xiang et al., 2020; Chang et al., 2015; Liu et al., 2022; Mo et al., 2019) as references. We select a total of 39 object categories, further divided into 134 sub-categories, and design generation rules for each. Based on the part connections and shapes observed in these datasets (Xiang et al., 2020; Chang et al., 2015; Liu et al., 2022; Mo et al., 2019), we define parameter sampling ranges for the advanced templates.

**Geometric Detail Synthesis Implementation.** We collect 524 objects spanning 39 categories from PartNet-Mobility (Xiang et al., 2020) and use our custom annotation system to derive the advanced templates and associated parameters that describe their spatial structures. We employ a frozen Dora (Chen et al., 2025) model as the point cloud encoder to extract features from both the point cloud of spatial structure and the reference point cloud with geometric details. The diffusion bridge is then trained using these features until convergence.

**Validation of Analytic Label Alignment.** Since the synthesized geometric details are not strictly aligned point-wise with the point clouds sampled from the synthesized spatial structure, some points may drift away from the mathematically defined regions. However, this drift is minimal. Specifically, the maximum distance between any point in the detailed point cloud and the underlying spatial structure[1] is bounded within 0.005 meters. This deviation is relatively small compared to the average object diameter of 0.86 meters. To mitigate this misalignment, we use this maximum drift (0.005 meters) as a threshold for determining whether a point belongs to a given region.

## A.2 EFFICIENCY AND COMPLEXITY OF SYNTHESIZED STRUCTURE OF ARTIPG++

**Synthesis Efficiency.** As discussed in Sec. 3.5, our pipeline requires only basic programming and math skills, allowing even first-year students to design procedural rules and knowledge annotations.

---

[1]The distance from a point to a spatial structure is defined as the minimum distance to any point sampled from the surface of the spatial structure.

Tab. 7 reports detailed statistics on the time required to create a novel object category, comparing Arti-PG with our method. Specifically, first-year college students with basic college-level mathematics knowledge were recruited to independently design procedural generation rules and knowledge descriptions (including semantics, instances, and affordances) for several object categories (including bottle, box, kettle, USB, washing machine, and storage furniture). Another group of volunteers with similar experience completed the same tasks using Arti-PG. For synthesis speed test, both methods were evaluated on identical hardware with Intel(R) Xeon(R) Gold 6133 CPU @ 2.25GHz and a single 4090 GPU. The comparison shows that creating procedural rules with ArtiPG++ is more efficient than with ArtiPG, especially achieving nearly 10 times faster object synthesis. **More importantly, we have packaged ArtiPG++ into a one-click tool for object synthesis and analytic label alignment, enabling the researchers to get free of manual effort and directly use ArtiPG++ to synthesize high-quality articulated objects and annotations.**

Table 7: Comparison of the time to create a novel object category. All values are in hours, except the synthesis time per object in seconds.

| Method | Defining Templates (hrs) | Defining Procedural Rules (hrs) | Manually Annotating External Assets (hrs) | Total Manual Effort (hrs) | Synthesis Speed (sec/obj) |
|--------|--------------------------|----------------------------------|--------------------------------------------|----------------------------|----------------------------|
| Arti-PG | 1.8 | 10.3 | 5.2 | 17.3 | 0.168 |
| Ours | 2.1 | 8.7 | 0.0 | 10.8 | 0.015 |

**Complexity of Synthesized Structure.** Here, we quantitatively evaluate the complexity of synthesized objects using our method. For each of the 39 object categories supported by our method, we randomly synthesize 10,000 instances and count the number of joints in each object to reflect the average structural complexity. We summarize the results by the frequency of objects with 1–9 joints and those with 10 or more joints, as shown in Fig. 4. From the joint count frequency distribution, it can be observed that these 39 categories have an average of 4 joints per object, comparable to the average in PartNet-Mobility (Xiang et al., 2020). For commonly complex objects such as *Dishwasher*, *Faucet*, *Oven*, *Window*, and *Chair*, the median number of joints reaches 6 or more. In addition, for *Storage Furniture*, about 20% of objects have 10 or more joints. These results demonstrate that our method synthesizes articulated objects with high structural complexity.

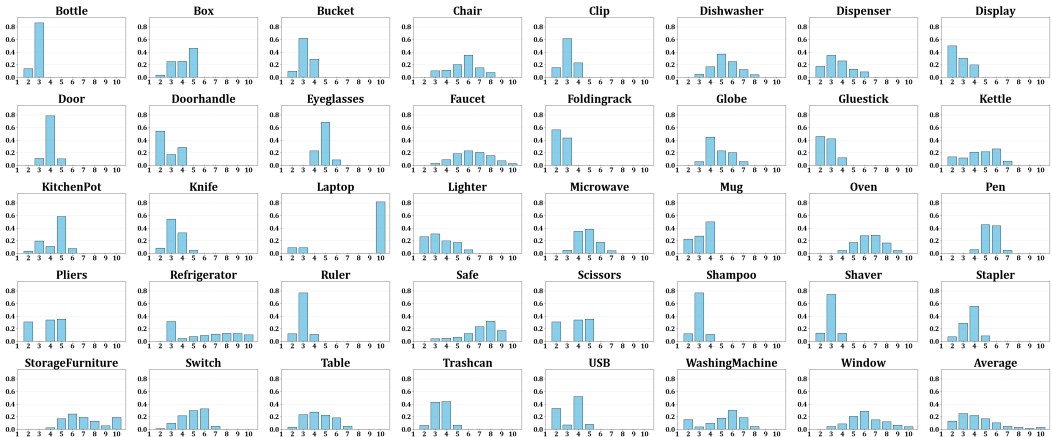

Figure 4: Statistics of synthesized joint counts. Each category is based on 10,000 randomly synthesized objects.

## A.3 MORE DETAILS ON EXPERIMENTS

### A.3.1 DIVERSITY

We discuss about the diversity of our synthesized objects in terms of both the spatial structure and the geometric details. In this section we provide detailed experimental results and analysis on the diversity.

**Discussion: Diversity in Spatial Structure.** The diversity of spatial structures in our synthesized articulated objects arises from two main aspects: 1) the variety of analytic templates, and 2) the diversity of the procedural structure synthesis process.

For the first aspect, we introduce analytic templates to describe the spatial structure of object components. Each template defines a family of geometry shapes sharing common structural properties, with specific instances generated by varying template parameters (*e.g.* radius, height, arc-angle). This parameterization enables the synthesis of a wide range of shapes from a single template. So far we have developed over 200 analytic templates, which can successfully model thousands of articulated objects across dozens of categories, capturing real-world diversity and structural complexity (Xiang et al., 2020).

For the second aspect, we design procedural rules that drive the synthesis of object structures at multiple levels. 1) **Parameter variation at the part level**: Templates incorporate randomized but valid parameters corresponding to shapes (*e.g.*, radius, height, arc-angle) to synthesize diverse instances of parts. 2) **Component duplication variation at the part level**: Templates support procedural duplication of certain components (*e.g.*, number of legs, number of vertical bars), introducing structural diversity within parts. 3) **Component substitution at the object level**: Individual object components can be instantiated using different advanced templates (*e.g.*, a handle may be synthesized as a "straight-handle", "curved-handle", or "round-handle"), enhancing diversity in overall object configurations. The combination of a large, expressive template set and flexible, randomized procedural synthesis rules enables ArtiPG++ to synthesize articulated objects with rich structural diversity within individual categories and across different types of objects.

**Discussion: Diversity in Geometric Details.** The diversity of geometric details in our synthesized articulated objects arises from two main aspects: 1) the diversity of reference objects used by the geometric detail synthesis, and 2) the influence of spatial structure on the geometric detail synthesis.

For the first aspect, we collect a wide range of articulated objects as reference data to train the diffusion bridge synthesizing geometric details as described in Sec. 3.3. These reference objects can be flexibly selected from various existing articulated object datasets. As a result, the diversity of geometric details in our synthesized objects is inherently broader than that of any single existing dataset.

For the second aspect, the diversity introduced by the spatial structure synthesis further enhances the diversity of geometric details. 1) Parameter variation at the part level causes the geometric details to adapt dynamically to variations in shape parameters, resulting in diverse surface features and contours. 2) Component duplication increases the surface area and structural regions available for detail generation, enabling richer diversity of fine-grained features across the object. 3) Component substitution at the object level allows different advanced templates to be instantiated for the same functional part, supporting a wide array of geometric detail combinations. These aspects enable our method to synthesize articulated objects with significantly rich and diverse geometric details.

Table 8: Per category evaluations of diversity. 'Real' denotes evaluation within real sets, and 'Syn' denotes evaluation within synthesized sets. The results are averaged over 20 independent runs.

| Metric | Method | Diversity Evaluation Results | | | | | | | | | |
|---|---|---|---|---|---|---|---|---|---|---|---|
| | | Bottle | Box | Bucket | Chair | Clip | Dishwasher | Dispenser | Display | Door | Doorhandle |
| | Real | 4.99 | 7.62 | 2.03 | 2.25 | 4.34 | 5.59 | 4.70 | 5.74 | 3.31 | 2.50 |
| | Syn | 5.26 | 8.76 | 6.18 | 3.04 | 5.62 | 6.30 | 5.51 | 6.06 | 3.67 | 2.63 |
| | | Eyeglasses | Faucet | Foldingrack | Globe | Gluestic | Kettle | Kitchenpot | Knife | Laptop | Lighter |
| | Real | 3.71 | 8.25 | 4.07 | 8.98 | 8.31 | 10.98 | 9.30 | 5.21 | 0.53 | 4.29 |
| NND($\times 10^{-4}$)↑ | Syn | 4.69 | 10.52 | 6.22 | 9.14 | 8.75 | 11.44 | 10.26 | 7.01 | 1.10 | 4.53 |
| | | Microwave | Mug | Oven | Pen | Pliers | Refrigerator | Ruler | Safe | Scissors | Shampoo |
| | Real | 9.14 | 11.34 | 9.32 | 1.70 | 3.69 | 6.41 | 3.24 | 9.69 | 3.43 | 7.83 |
| | Syn | 10.35 | 11.58 | 10.44 | 2.75 | 4.24 | 7.11 | 4.08 | 7.11 | 3.96 | 8.15 |
| | | Shaver | Stapler | Storage | Switch | Table | Trashcan | USB | Washing | Window | AVG |
| | Real | 4.52 | 12.64 | 11.85 | 4.78 | 12.55 | 8.14 | 1.21 | 8.77 | 1.68 | 6.12 |
| | Syn | 6.39 | 13.33 | 12.26 | 6.10 | 14.05 | 8.57 | 2.00 | 9.03 | 2.04 | 7.03 |

**Evaluation.** We use "nearest neighbor distance" (NND) to quantitatively evaluate the diversity of synthesized objects. For each of the 39 object categories, we collect real objects from PartNet-Mobility (Xiang et al., 2020) and PartNet (Mo et al., 2019) to form the real set $S_r$, and synthesize an equal number of objects to form the synthesized set $S_s$. All objects are aligned by centering their

bounding boxes at the origin, without applying any rotation. We then compute the NND between these sets as:

$$\text{NND}(S) = \frac{1}{|S|} \sum_{X \in S} \min_{Y} \left\{ \text{CD}(X, Y) | Y \in S - \{X\} \right\} \tag{1}$$

where CD is the Chamfer Distance.

Under the same set size, a higher NND value indicates greater diversity among objects. We report the results in Tab. 8. On average, the NND for real object sets is $6.12 \times 10^{-3}$, while our synthesized sets achieve $7.03 \times 10^{-3}$. This demonstrates that the diversity of our synthesized objects is not only comparable to, but even slightly exceeds that of real-world datasets.

### A.3.2 DATA DISTRIBUTION

Beyond diversity, it is crucial that the distribution of our synthesized objects aligns well with that of real objects. To evaluate this, we use Férchet Inception Distance (FID) (Heusel et al., 2017) as a quantitative metric. For each of the 39 object categories, we randomly split the real dataset (Mo et al., 2019; Xiang et al., 2020) into two equal sized subsets $S_{r1}$ and $S_{r2}$, and synthsize an object set $S_s$ of the same size, i.e., $|S_s| = |S_{r1}| = |S_{r2}|$.

Since FID relies on a feature extractor, we follow PointNet++ (Qi et al., 2017) to train a classifier on the 26 categories and use its encoder to extract feature vectors from point clouds.[2]

For each category, we compute the average FID between synthesized and real subsets, i.e. $\frac{\text{FID}(S_s, S_{r1}) + \text{FID}(S_s, S_{r2})}{2}$. As a baseline, we also calculate $\text{FID}(S_{r1}, S_{r2})$ between the two real subsets. We repeat the experiment 20 times and report the average in Tab. 9. The FID between synthesized and real sets is $4.55 \times 10^3$, while that between real subsets is $4.16 \times 10^3$. These results confirm that our synthesized objects exhibit a distribution closely matching that of real-world datasets.

Table 9: Per category evaluations of distribution. "Real-Real" denotes evaluation between two divided real sets of equal size, and "Syn-Real" denotes evaluation between synthesized sets and real sets. "$\Delta$" denotes the differences between the two distribution similarities. Less $\Delta$ represents the distribution similarity between synthesized objects and real-world objects is closer to the one among real-world objects. The results are averaged over 20 independent runs.

| Metric | Method | Distribution Evaluation Results | | | | | | | | | |
|---|---|---|---|---|---|---|---|---|---|---|---|
| | | Bottle | Box | Bucket | Chair | Clip | Dishwasher | Dispenser | Display | Door | Doorhandle |
| | Real-Real | 2.09 | 3.01 | 2.08 | 4.25 | 3.04 | 3.62 | 5.45 | 2.01 | 4.57 | 1.82 |
| | Syn-Real | 3.53 | 3.55 | 5.12 | 5.03 | 2.95 | 5.11 | 6.72 | 2.13 | 3.64 | 1.90 |
| | $\Delta$ | 1.44 | 0.54 | 3.04 | 0.78 | -0.09 | 1.49 | 1.27 | 0.12 | -0.93 | 0.08 |
| | | Eyeglasses | Faucet | Foldingrack | Globe | Gluestic | Kettle | Kitchenpot | Knife | Laptop | Lighter |
| | Real-Real | 2.02 | 5.31 | 2.31 | 4.56 | 5.26 | 4.61 | 3.97 | 4.33 | 2.00 | 2.27 |
| FID($\times 10^3$) | Syn-Real | 2.20 | 6.19 | 2.45 | 4.68 | 6.49 | 3.32 | 3.57 | 4.50 | 2.22 | 2.92 |
| | $\Delta$ | 0.18 | 0.88 | 0.14 | 0.12 | 1.23 | -1.29 | -0.40 | 0.17 | 0.22 | 0.65 |
| | | Microwave | Mug | Oven | Pen | Pliers | Refrigerator | Ruler | Safe | Scissors | Shampoo |
| | Real-Real | 6.46 | 6.12 | 5.30 | 6.38 | 2.48 | 3.11 | 2.35 | 4.83 | 3.07 | 5.31 |
| | Syn-Real | 8.19 | 6.87 | 5.07 | 7.37 | 2.89 | 3.59 | 2.92 | 4.17 | 3.48 | 5.67 |
| | $\Delta$ | 1.73 | 0.75 | -0.23 | 0.99 | 0.41 | 0.48 | 0.57 | -0.66 | 0.41 | 0.36 |
| | | Shaver | Stapler | Storage | Switch | Table | Trashcan | USB | Washing | Window | AVG |
| | Real-Real | 5.55 | 8.73 | 9.13 | 2.56 | 9.92 | 2.01 | 6.04 | 2.11 | 2.39 | 4.16 |
| | Syn-Real | 4.82 | 8.83 | 8.93 | 2.71 | 10.30 | 2.09 | 6.38 | 2.17 | 2.67 | 4.55 |
| | $\Delta$ | -0.73 | 0.10 | -0.20 | 0.15 | 0.38 | 0.08 | 0.34 | 0.06 | 0.28 | 0.38 |

Table 10: Detailed statistics of the data split on part segmentation task. The number of training / testing objects across 26 categories are involved in the tasks.

| Training / Testing Objects for Segmentation Task | | | | | | | | |
|---|---|---|---|---|---|---|---|---|
| Bottle | Box | Bucket | Chair | Dishwasher | Dispenser | Display | Door | Eyeglasses |
| 64 / 400 | 18 / 10 | 18 / 18 | 27 / 54 | 12 / 36 | 19 / 38 | 50 / 904 | 24 / 12 | 43 / 22 |
| Globe | Kettle | KitchenPot | Laptop | Lighter | Microwave | Pen | Pliers | Refrigerator |
| 40 / 20 | 18 / 10 | 15 / 10 | 48 / 405 | 18 / 10 | 6 / 10 | 32 / 16 | 10 / 14 | 30 / 14 |
| Safe | Scissors | Stapler | Switch | TrashCan | USB | Washing | Window | |
| 20 / 10 | 32 / 15 | 13 / 10 | 47 / 23 | 37 / 19 | 20 / 31 | 7 / 10 | 35 / 18 | |

---

[2]The encoder consists of four 3D convolutional layers followed by max-pooling and an MLP, taking as input a point cloud of 2048 sampled points.

Table 11: Detailed statistics of the data split on point cloud completion tasks. The number of training / testing objects across 39 categories are involved in the task.

| Training / Testing Objects for **Point Cloud Completion** Task | | | | | | | | | | | | |
|---|---|---|---|---|---|---|---|---|---|---|---|---|
| Bottle
64 / 400 | Box
18 / 10 | Bucket
18 / 18 | Chair
27 / 54 | Clip
13 / 36 | Dishwasher
12 / 36 | Dispenser
19 / 38 | Display
50 / 904 | Door
24 / 12 | Doorhandle
5 / 10 | Eyeglasses
43 / 22 | Faucet
50 / 200 | Foldingrack
2 / 6 |
| Globe
40 / 20 | Gluestick
12 / 32 | Kettle
18 / 10 | KitchenPot
15 / 10 | Knife
22 / 54 | Laptop
48 / 405 | Lighter
18 / 10 | Microwave
6 / 10 | Mug
30 / 83 | Oven
6 / 12 | Pen
32 / 16 | Pliers
10 / 14 | Refrigerator
30 / 14 |
| Ruler
5 / 10 | Safe
20 / 10 | Scissors
32 / 15 | Shampoo
12 / 34 | Shaver
3 / 6 | Stapler
13 / 10 | Storage
137 / 500 | Switch
36 / 100 | Table
47 / 23 | TrashCan
37 / 19 | USB
20 / 31 | Washing
7 / 10 | Window
35 / 18 |

### A.3.3 VISION TASKS

**Vision Experiment Settings.** Here we provide more details for vision tasks settings.

For the part segmentation task, we use Point-Transformer (Zhao et al., 2021) as the baseline. Each object is represented as a point cloud of 2,048 points uniformly sampled from the mesh, with point-wise part segmentation annotations. We follow a category-wise train/test split detailed in Tab. 10. The model is trained for 200 epochs using the Adam optimizer with a learning rate of 0.001, weight decay of 0.0001, and a batch size of 32. The commonly used mean accuracy and mean IoU are adopted as the metrics.

For the point cloud completion task, we adopt SnowflakeNet (Xiang et al., 2021) as the baseline model. The data split is reported in Tab. 11. Each training sample consists of an incomplete point cloud with 2,048 points and a corresponding complete point cloud with 16,384 points. We use the official SnowflakeNet implementation with four cascaded upsampling stages. The model is trained for 150 epochs using the Adam optimizer with an learning rate of 0.0001 and a batch size of 32. Chamfer Distance is used as the evaluation metric.

For the part pose estimation task, we follow GAPartNet (Geng et al., 2023) for data preparation. Specifically, we render RGB-D images of articulated objects in SAPIEN simulator (Xiang et al., 2020) with annotations, variate collected data by using random camera poses and joint poses and finally gather 20000 points as input. The position and orientation of parts are defined in the Normalized Part Coordinate Space (NPCS). Specifically, each detectable part is reduced to a standard orientation and normalized within a unit ball. We use batch sizes of 64, depending on the default settings of baseline models. We use Adam optimizer with learning rate of 0.001 and weight decay of 0.0001 to optimize the network parameters.

Table 12: Detailed experimental results on part segmentation task.

| Metric | Method | Part Segmentation Results | | | | | | | | |
|---|---|---|---|---|---|---|---|---|---|---|
| | | Bottle | Box | Bucket | Chair | Dishwasher | Dispenser | Display | Door | Eyeglasses |
| **mAcc**(%) ↑ | Real Obj | 95.4 | 95.4 | 96.3 | 92.5 | 88.1 | 90.9 | 93.9 | 77.9 | 96.5 |
| | Arti-PG | 96.6 | 97.2 | 98.5 | 94.5 | 90.2 | 91.3 | 96.4 | 78.1 | 97.1 |
| | Ours | 97.2 | 97.5 | 98.5 | 94.7 | 90.5 | 92.0 | 96.6 | 80.2 | 97.1 |
| | | Globe | Kettle | KitchenPot | Laptop | Lighter | Microwave | Pen | Pliers | Refrigerator |
| | Real Obj | 95.9 | 89.7 | 90.3 | 96.8 | 92.3 | 82.8 | 85.7 | 74.0 | 94.1 |
| | Arti-PG | 96.9 | 93.9 | 95.8 | 97.1 | 93.8 | 90.3 | 87.6 | 75.2 | 94.3 |
| | Ours | 97.2 | 94.5 | 96.0 | 98.0 | 94.3 | 90.5 | 87.9 | 75.8 | 94.4 |
| | | Safe | Scissors | Stapler | Switch | TrashCan | USB | Washing | Window | AVG |
| | Real Obj | 92.8 | 90.5 | 79.9 | 84.5 | 92.2 | 82.6 | 91.6 | 87.2 | 89.5 |
| | Arti-PG | 94.6 | 90.6 | 82.8 | 85.0 | 92.7 | 82.8 | 92.1 | 91.6 | 91.3 |
| | Ours | 95.2 | 90.8 | 85.3 | 85.5 | 92.8 | 83.3 | 92.4 | 92.4 | 91.8 |
| | | Bottle | Box | Bucket | Chair | Dishwasher | Dispenser | Display | Door | Eyeglasses |
| **mIoU**(%) ↑ | Real Obj | 75.1 | 93.7 | 48.5 | 78.2 | 60.5 | 78.2 | 81.9 | 52.1 | 92.9 |
| | Arti-PG | 82.7 | 96.4 | 49.8 | 80.1 | 65.4 | 78.5 | 84.0 | 56.0 | 94.1 |
| | Ours | 83.0 | 96.8 | 62.1 | 81.0 | 67.2 | 80.0 | 84.5 | 61.5 | 94.4 |
| | | Globe | Kettle | KitchenPot | Laptop | Lighter | Microwave | Pen | Pliers | Refrigerator |
| | Real Obj | 85.5 | 88.7 | 92.8 | 87.7 | 72.5 | 74.5 | 65.9 | 75.6 | 61.3 |
| | Arti-PG | 94.2 | 93.5 | 97.6 | 88.8 | 84.1 | 81.1 | 66.6 | 88.5 | 64.9 |
| | Ours | 95.1 | 94.9 | 97.6 | 90.9 | 84.7 | 82.0 | 72.3 | 89.9 | 69.2 |
| | | Safe | Scissors | Stapler | Switch | TrashCan | USB | Washing | Window | AVG |
| | Real Obj | 86.8 | 56.5 | 74.3 | 71.0 | 72.7 | 87.6 | 48.2 | 68.2 | 74.5 |
| | Arti-PG | 89.0 | 61.5 | 83.5 | 71.8 | 82.9 | 88.6 | 53.3 | 73.0 | 79.4 |
| | Ours | 91.2 | 63.4 | 83.8 | 74.0 | 83.0 | 88.8 | 59.7 | 75.2 | 81.3 |

**More Vision Task Results.** We present detailed results for part segmentation and point cloud completion across all object categories in Tab. 12 and Tab. 13, respectively. As part pose estimation is not a category-specific task, we do not procide categorical results. Across both tasks and all baseline methods, our approach consistently demonstrates significant improvements in performance for each object category. This strong performance is largely attributed to the ability of our method to synthesize articulated structures with extensive diversity while maintaining physical plausibility. These results offer more comprehensive and compelling evidence of the effectiveness and superiority of our proposed approach.

Table 13: Detailed experimental results on point cloud completion task. "-" means Arti-PG does not contain the category.

| Metric | Method | Point Cloud Completion Results | | | | | | | | | |
|---|---|---|---|---|---|---|---|---|---|---|---|
| | | Bottle | Box | Bucket | Chair | Clip | Dishwasher | Dispenser | Display | Door | Doorhandle |
| | Real Obj | 9.7 | 14.6 | 14.4 | 8.2 | 10.3 | 12.2 | 14.1 | 9.4 | 8.6 | 6.5 |
| | Arti-PG | 9.6 | 14.0 | 13.0 | - | - | - | - | 9.3 | 8.5 | - |
| | Ours | 8.2 | 12.1 | 12.8 | 6.5 | 9.1 | 10.8 | 13.0 | 9.0 | 8.5 | 6.5 |
| | | Eyeglasses | Faucet | Foldingrack | Globe | Gluestic | Kettle | Kitchenpot | Knife | Laptop | Lighter |
| | Real Obj | 5.1 | 9.0 | 9.1 | 18.2 | 8.8 | 19.4 | 17.6 | 9.2 | 9.4 | 8.6 |
| | Arti-PG | 5.1 | - | - | 17.0 | - | 19.0 | 16.4 | - | 7.1 | 7.2 |
| CD($\times 10^{-4}$cm)$\downarrow$ | Ours | 4.9 | 8.8 | 7.0 | 13.3 | 8.6 | 13.3 | 12.2 | 9.0 | 7.0 | 6.2 |
| | | Microwave | Mug | Oven | Pen | Pliers | Refrigerator | Ruler | Safe | Scissors | Shampoo |
| | Real Obj | 15.8 | 7.2 | 14.7 | 4.7 | 6.5 | 8.9 | 6.9 | 15.2 | 5.0 | 12.3 |
| | Arti-PG | 13.3 | - | - | 4.7 | 5.3 | 8.8 | - | 12.2 | 4.6 | - |
| | Ours | 10.9 | 6.8 | 12.5 | 4.3 | 5.1 | 6.9 | 6.5 | 10.7 | 4.6 | 9.7 |
| | | Shaver | Stapler | Storage | Switch | Table | Trashcan | USB | Washing | Window | AVG |
| | Real Obj | 11.5 | 9.6 | 21.5 | 13.6 | 19.8 | 12.7 | 8.9 | 16.2 | 6.9 | 11.3 |
| | Arti-PG | - | 8.4 | - | 13.5 | - | 12.0 | 8.0 | 16.0 | 5.3 | 10.4 |
| | Ours | 10.8 | 8.1 | 16.3 | 10.3 | 13.5 | 11.8 | 8.0 | 11.3 | 5.2 | 9.2 |

Table 14: List of specific tasks in manipulation.

| Category | Tasks |
|---|---|
| Box | Lift Lid |
| Bucket | Move Handle |
| Door | Open/Close Door; Open Door via Handle |
| Faucet | Turn on/off Switch |
| Fridge | Open/Close Door; Open Door via Handle |
| Kettle | Move Handle |
| KitchenPot | Lift Lid |
| Microwave | Open/Close Door; Open Door via Handle |
| Safe | Open/Close Door; Open Door via Handle |
| Storage | Open/Close Door; Open Door via Handle; Open Drawer via Handle |
| Switch | Turn on/off Switch |
| Table | Open/Close Door; Open Door via Handle; Open Drawer via Handle |
| TrashCan | Lift Lid |
| WashingMachine | Open/Close Door; Open Door via Handle; Lift Lid |
| Window | Open/Close Window; Open Window via Handle |

### A.3.4 MANIPULATION TASKS

**Manipulation Experiment Settings.** As shown in Tab. 14, we conduct manipulation experiments on 15 representative object categories and tasks. To focus evaluation on the understanding of articulated object structures and affordance detection, we exclude objects that are either too small (*e.g.*, Pen, USB) or unsuitable for single-gripper manipulation (*e.g.*, Bottle, Scissors), following the setup in Where2Act (Mo et al., 2021).

Following the practice of Where2Act (Mo et al., 2021) and ManipLLM (Li et al., 2024), We use the SAPIEN (Xiang et al., 2020) simulator as the interaction environment. Each simulation begins with an object placed at the center of the scene, with its joint pose randomly set to a closed or partially open configuration (50% each). The scene is captured by an RGB-D camera with known intrinsics, positioned randomly on the upper hemisphere with azimuth $\in [0°, 360°)$ and altitude $\in [30°, 60°]$, and directed at the object center. A Franka Panda Flying gripper with two fingers is used for interaction. The action proposals are based on the affordances. Specifically, the gripper is placed 0.05 m from the target along the movement direction, moves forward to push or grasp it, and retracts if the task involves pulling or lifting the target part.

For the classic frameworks Where2Act (Mo et al., 2021) and Where2Explore (Ning et al., 2023), we follow their approach to select the pixel with the highest predicted action likelihood and use the

corresponding gripper orientation and movement direction from the action proposal. Real-object affordances are collected in simulation, while synthesized-object affordances are automatically assigned using region-based analytic knowledge alignment as described in Sec. 3.4. For GAPartNet (Geng et al., 2023), we detect actionable parts and estimate their poses to derive the gripper orientation and motion. Affordances for real objects follow NPCS annotations and they are derived using pose-based analytic knowledge alignment for synthesized objects. For ManipLLM (Li et al., 2024), affordances are inferred by prompting the LLM with RGB images. Real-object affordances are collected in simulation, and those for synthesized objects are assigned via region-based analytic knowledge. All methods are trained following their original procedures and hyperparameters.

Tab. 15 lists the specific training and testing categories of the 15 object categories. Specifially, five categories are selected as testing categories with no training objects to evaluate the generalizability of the frameworks trained with our synthesized objects.

Table 15: Detailed statistics of the data split on manipulation tasks. "-" for training objects means the category is unseen in training.

| Type | Training / Testing Objects for **Manipulation** Tasks | | | | |
|---|---|---|---|---|---|
| Training Categories | Box
20 / 8 | Door
23 / 12 | Faucet
65 / 19 | Kettle
22 / 7 | Microwave
9 / 3 |
| | Fridge
32 / 11 | Storage
270 / 75 | Switch
53 / 17 | TrashCan
52 / 17 | Window
40 / 18 |
| Testing Categories | Bucket
- / 36 | KitchenPot
- / 23 | Safe
- / 29 | Table
- / 95 | Washing
- / 16 |

**Detailed Manipulation Results.** We present detailed manipulation results by object category in Tab. 16. In most cases, neural networks trained on our synthesized objects outperform those trained on other objects. Since manipulation tasks require comprehensive understanding of spatial structures and functional affordances, these results demonstrate the effectiveness of our method in synthesizing high-quality articulated objects with analytically precise annotations.

Table 16: Per category experimental results on manipulation. All values are percentage sample success rate.

| Network | Data | **Object Manipulation** Results | | | | | | | | | | | | | | | | |
| | | Training Categories | | | | | | | | | | | Testing Categories | | | | | |
| | | Box | Dor | Fct | Fdr | Ket | Mcw | Stf | Swt | Tcn | Win | AVG | Bkt | Pot | Saf | Tab | Wsm | AVG |
|---|---|---|---|---|---|---|---|---|---|---|---|---|---|---|---|---|---|---|
| W2A | Real | 16.8 | 31.5 | 17.1 | 31.2 | 9.9 | 31.3 | 37.1 | 19.3 | 17.1 | 12.2 | 26.1 | 8.0 | 6.2 | 13.4 | 17.8 | 9.1 | 14.4 |
| | Arti-PG | 27.0 | 31.9 | 24.3 | 31.1 | 23.7 | 34.2 | 30.7 | 26.3 | 16.7 | 16.2 | 26.7 | 11.9 | 6.9 | 16.7 | 26.8 | 17.0 | 16.9 |
| | Ours | 29.3 | 34.2 | 26.6 | 33.4 | 26.0 | 36.5 | 33.0 | 28.6 | 19.0 | 18.5 | 29.0 | 14.3 | 7.1 | 17.2 | 28.5 | 17.9 | 20.8 |
| W2E | Real | 19.0 | 38.3 | 16.8 | 35.4 | 11.5 | 33.8 | 34.6 | 23.6 | 19.5 | 15.8 | 26.9 | 14.0 | 10.8 | 24.8 | 22.7 | 15.4 | 20.5 |
| | Arti-PG | 29.3 | 39.8 | 23.7 | 35.9 | 18.4 | 35.0 | 36.7 | 26.4 | 29.6 | 18.0 | 28.0 | 20.0 | 15.4 | 25.5 | 34.6 | 19.3 | 25.7 |
| | Ours | 33.4 | 40.9 | 27.8 | 40.0 | 22.5 | 37.1 | 35.8 | 28.5 | 30.7 | 18.1 | 32.1 | 24.6 | 18.4 | 26.5 | 38.0 | 22.1 | 30.2 |
| GA | Real | 25.5 | 40.4 | 17.1 | 40.5 | 11.1 | 32.8 | 40.5 | 18.2 | 19.5 | 13.8 | 29.7 | 23.3 | 10.7 | 25.2 | 30.9 | 12.7 | 29.0 |
| | Arti-PG | 37.0 | 45.3 | 25.6 | 46.2 | 24.2 | 35.0 | 36.0 | 26.7 | 26.7 | 19.3 | 32.8 | 26.1 | 24.2 | 25.4 | 39.8 | 19.5 | 30.1 |
| | Ours | 40.2 | 48.7 | 28.0 | 49.9 | 27.1 | 37.3 | 39.7 | 29.1 | 29.2 | 22.3 | 35.7 | 28.1 | 22.2 | 32.4 | 41.5 | 25.5 | 34.2 |
| ManipLLM | Real | 31.0 | 42.4 | 19.4 | 42.8 | 13.2 | 36.7 | 41.8 | 25.5 | 20.7 | 14.5 | 32.0 | 19.9 | 12.4 | 26.8 | 32.3 | 18.4 | 30.6 |
| | Arti-PG | 45.7 | 50.2 | 20.5 | 45.4 | 15.6 | 41.5 | 44.0 | 28.4 | 25.0 | 20.7 | 33.5 | 28.9 | 26.8 | 29.0 | 40.1 | 20.1 | 31.6 |
| | Ours | 46.2 | 51.9 | 27.1 | 48.0 | 29.7 | 41.8 | 45.2 | 29.4 | 31.4 | 23.7 | 36.4 | 31.9 | 29.7 | 39.5 | 42.3 | 32.8 | 36.1 |

### A.3.5 DETAILS FOR ABLATION STUDY

**Feature Extractor in Geometric Detail Synthesis.** In Sec. 4.4, we briefly discussed the ablation study on feature extractors in the diffusion bridge for geometric detail synthesis, including downstream articulated object understanding tasks using synthesized objects with PointNet++ (Qi et al., 2017), Craftsman VAE (Li et al., 2025), and our implementation Dora (Chen et al., 2025). The quantitative results are reported in Tab. 6. Here, we provide visualizations of the geometric details synthesized by these extractors and examine their influence on annotations through the same analytic label alignment process, to further analyze their impact on downstream tasks.

As shown in Fig. 5, geometric details generated by PointNet++ (Qi et al., 2017) and Craftsman (Li et al., 2025) often misalign with the spatial structure, especially at fine-grained articulation joints. For example, in faucet, the spatial structure (*Faucet-(a)*) defines five handles per knob, which

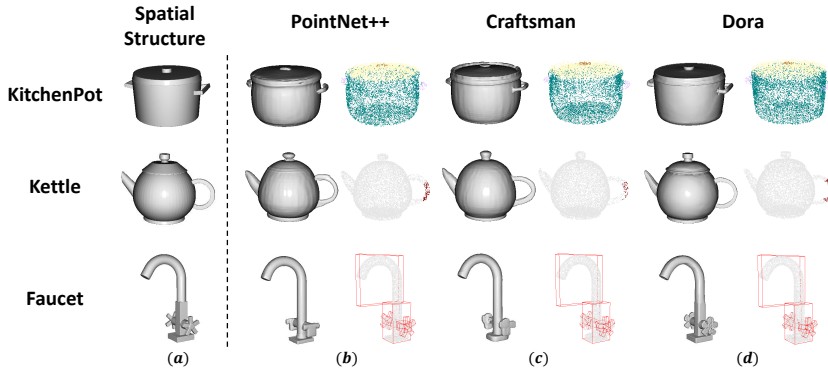

Figure 5: Visualization of objects with geometric details and annotations. Part segmentation labels are shown for the kitchen pot, affordance labels for the kettle, and bounding boxes indicating part poses for the faucet.

Dora (Chen et al., 2025) preserves with realistic details (*Faucet-(d)*), while PointNet++ (Qi et al., 2017) and Craftsman (Li et al., 2025) distort the handles into unrecognizable shapes (*Faucet-(b),(c)*).

Such misalignment in spatial structure and geometric details further disrupts analytic label alignment. Craftsman (Li et al., 2025) generates a kitchen pot lid with extra segmentation layers (*KitchenPot-(c)*) and a kettle handle with curve (*Kettle-(c)*) misaligned with the structure (*Kettle-(a)*), leading to incomplete affordance label alignment. For part pose, the issue is even clearer that handles generated by PointNet++ (Qi et al., 2017) and Craftsman (Li et al., 2025) cannot align with the point cloud (*Faucet-(b),(c)*), resulting in poor downstream task accuracy. In contrast, Dora (Chen et al., 2025) maintains realism while closely adhering to the spatial structure, enabling accurate analytic label alignment and effective downstream training (*Faucet-(d)*).

## A.4 REAL-WORLD EXPERIMENT DEMONSTRATIONS

**We provide video demonstrations of real-world experiments in the *experiment-videos* folder in our supplementary material**, showcasing manipulation tasks on various articulated objects, including a bucket, door, microwave, kitchen pot, and storage furniture. The experiments are conducted using a Flexiv Rizon robotic arm with a parallel gripper and an Intel RealSense D415 camera.

## A.5 IMPLEMENTATION EXAMPLES

In this section, we provide the implementation of the advanced templates program and the procedural rules to synthesize spatial structure in Python and provide detailed explanations. We use "*Bucket*" as an example. We omit certain codes for simplicity such as "importing packages". **We also provide code examples for "*Chair*", "*Kettle*", and "*Microwave*" in the *code-examples* folder in our supplementary material.** Our codes for synthesizing all 39 categories of objects will be publicly available.

### A.5.1 ADVANCED TEMPLATES

**Base Template.** First, we implement the base class for elementary primitives. It mainly contains the offset and rotation of an elementary primitive. The elementary primitive can be further moved in 3D space through functions like `translation` and `rotation`.

```python
class Base_Template:
  def __init__(
    self,
    translation = [0,0,0],
    rotation = [0,0,0],
  ):
    """
```

```
      :param offset: pose parameters for the elementary primitive's
          initial position.
      :param rotation: pose parameters for the elementary primitive's
          initial rotation in Euler angles.
      """
      self.offset = offset
      self.rotation = rotation
      self.structure = None

  def translate(self, translation):
      """
      Translate the primitive according to the given values.
      """
      self.structure.translate(offset)

  def rotate(self, rotation):
      """
      Rotate the primitive (around the origin) according to the given
          values.
      """
      self.structure.rotate(rotation)
```

**Geometry Template.**   We show the codes for class `Cylinder` as an geometry template that represent basic geometry shapes. During initialization, it registers the parameters $R, h$ and creates a mesh of the cylinder.

```
class Cylinder(Elementary_Primitive):
  def __init__(
    self, R, h,
    translation=[0,0,0],
    rotation=[0,0,0]
  ):
      """
      :param R: radius of the cylinder
      :param h: height of the cylinder
      :param offset: offset (x, y, z) of the cylinder
      :param rotation: rotation of the cylinder, represented via Euler
          angles (x, y, z)
      """
      super().__init__(offset, rotation)
      self.R = R
      self.h = h
      self.structure = create_mesh(
          'cylinder',
          radius=R, height=h,
          offset=offset,
          rotation=rotation
      )
```

**Advanced Template.**   Below we give the implementation of a specific advanced template, *i.e.* the main body of a bucket with common cylindrical shape.

```
class Cylindrical_Body():
  default_parameters = {
    'outer_sizes': ...,
    'inner_sizes': ...,
    ...
  }
  def __init__(self,
    outer_sizes, inner_size,
    translation=[0, 0, 0], rotation=[0, 0, 0]
  ):
```

```
1134
1135        self.outer_sizes = outer_sizes
1136        self.inner_sizes = inner_sizes
1137        self.translation = translation
1138        self.rotation = rotation
1139
1140        '''
             Define the components of bucket body.
1141        '''
1142        self.bottom = Cylinder(
1143          outer_sizes[2]-inner_sizes[2],
             outer_size[0]*(1-inner_size[2]/outer_size[2])+ \
1144             outer_size[1]*inner_size[2]/outer_size[2],
1145          outer_size[1],
1146          position=[0,-inner_size[2]/2,0]
1147        )
1148        self.body = Ring(
1149          inner_size[2],
             outer_size[0],
1150          inner_size[0],
1151          position=[0,(outer_size[2]-inner_size[2])/2,0]
1152        )
1153
1154        '''
             Define the connection type between the bottom and main frastructure
1155             of bucket body.
1156        '''
1157        self.connect(self.bottom, self.body, type='fixed')
1158
1159
```

### A.5.2 PROCEDURALLY GENERATION OF SPATIAL STRUCTURE

The procedural geneation processes for each of the 39 categories are encapsuled into a Python class for easy use.

```
class Bucket:
  def __init__(self):
    self.category = 'Bucket'
    self.default_parameters = {
       'body_sizes': ...,
       'handle_sizes': ...,
       ...
    }
```

**Determine Category & Components.** Here we determine the specific category of the synthesized object and the components of it. Each of the 39 object categories supported by our method is divided into several sub-categories to distinguish variations in shape and functionality for subsequent procedural generation of spatial structure.

```
def determine_category(self):
  sub_categories = [
    'Cylindrical_Bucket',
    'Prism_Bucket',
    ...
  ]
  self.sub_category = random.choice(sub_categories)

def determine_components(self):
  available_templates = {
    'body': ['Cylindrical_Body', 'Prism_Body', ...],
    'handle': ['Round_Handle', 'Curved_Handle', ...],
    'cover: ['Cylindrical_Cover', 'Arc_Cover', ...]
  }
```

```
1188
1189    chosen_templates = []
1190    for component, templates in available_templates:
1191      if self.sub_category == ...:
1192        chosen_templates.append(random.choice(templates)
1193      else:
1194        ...
1195
1196    self.chosen_templates = chosen_templates
1197
```

**Determine Parameters & Instantiate Components.** As described in Sec. 3.2, we model each object as a tree, where the root node represents the fundamental part. We then follow a pre-order traversal order to define the shape of each part by filling in parameters for its corresponding template.

```
def determine_parameters(self):
  for template in templates:
    parameters = template.default_parameters

    '''
    Traverse all components in a pre-defined order and determine their
        parameters.
    '''

    if template == 'Cylindrical_Body':
      parameters['outer_size'] = stochastic_parameters(...)
      parameters['inner_size'] = stochastic_parameters(...)
      parameters['position'] = stochastic_parameters(...)
      parameters['rotation'] = stochastic_parameters(...)
      self.cylindrical_body = eval('Cylindrical_Body')(parameters)

    elif template == 'Round_Handle':
      ...

    else:
      ...
```

## A.6 TEXTURE SYNTHESIS

Despite the core contribution of our work is to propose an efficient way in synthesizing artiuclated objects and collection high-quality annotations, texture synthesis is essential for synthesizing diverse, realistic, and high-quality 3D objects. After synthesizing the spatial structure and geometric details of an articulated object, it can be exported as a mesh. We follow Paint-It (Youwang et al., 2024) to apply realistic and diverse textures to the mesh. Given the object category, multi-view images of the mesh, and the object mesh itself, Paint-It can generate category-appropriate textures and accurately map them onto the mesh surface. In Fig. 6, we showcase examples of synthesized objects with textures, including a laptop, bottle, USB, kitchen pot, scissors, kettle, and faucet.

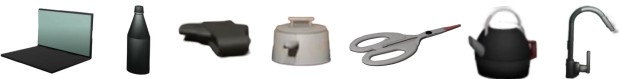

Figure 6: Illustration of synthesized objects with texture.

## A.7 QUALITATIVE COMPARISON WITH EXISTING METHODS

We provide a qualitative comparison with other articulated object synthesis methods, including NAP (Lei et al., 2023), CAGE (Liu et al., 2024), and the state-of-the-art MeshArt (Gao et al., 2025). The visualization results are shown in Fig. 7. NAP (Lei et al., 2023) requires Dual Contouring (Ju et al., 2002) to decode geometries from synthesized shapes, which we implement with Chen et al.

(2022). This results in overly tessellated object surfaces, whereas our method produces smooth, continuous surfaces with recognizable geometric details. Additionally, for CAGE (Liu et al., 2024) and MeshArt Gao et al. (2025), the synthesized articulated objects are often composed of regular geometric primitives, lacking irregular geometric details commonly found in real-world objects. In contrast, our method incorporates such geometric details into the synthesized articulated objects, making them more realistic.

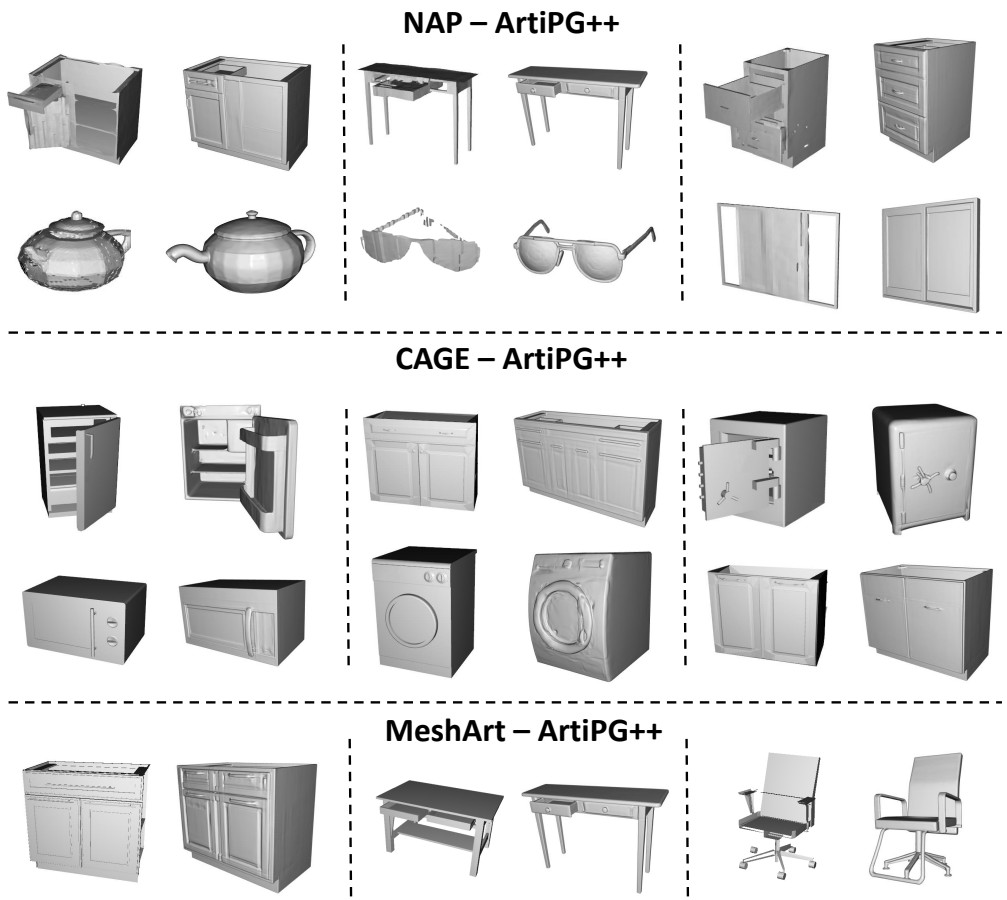

Figure 7: Visualization of qualitative comparison with existing methods. In each pair shown, the left image is synthesized by a baseline method, and the right image is synthesized by our method.

## A.8    VISUALIZATION OF SYNTHESIZED OBJECTS

In this section, we provide visualizatin of synthesized objects for qualitative evaluation in Fig. 8. We provide 32 categories, including bottle, box, bucket, chair, clip, dishwasher, dispenser, door, eyeglasses, faucet, glove, kettle, kitchenpot, knife, laptop, microwave, mug, oven, pen, pliers, refrigerator, ruler, safe, shampoo, stapler, storagefurniture, switch, table, trashcan, usb, washing machine, and window. Each category contains four objects, rendered in white model without texture for better demonstrate their structures.

## A.9    STATEMENT ON USAGE OF LLMS

In this work, LLMs were used only for proofreading, polishing the manuscript, and assisting in writing the codes of program templates and procedural rules. All scientific ideas, experimental designs, results, and conclusions are entirely the original work of the authors.

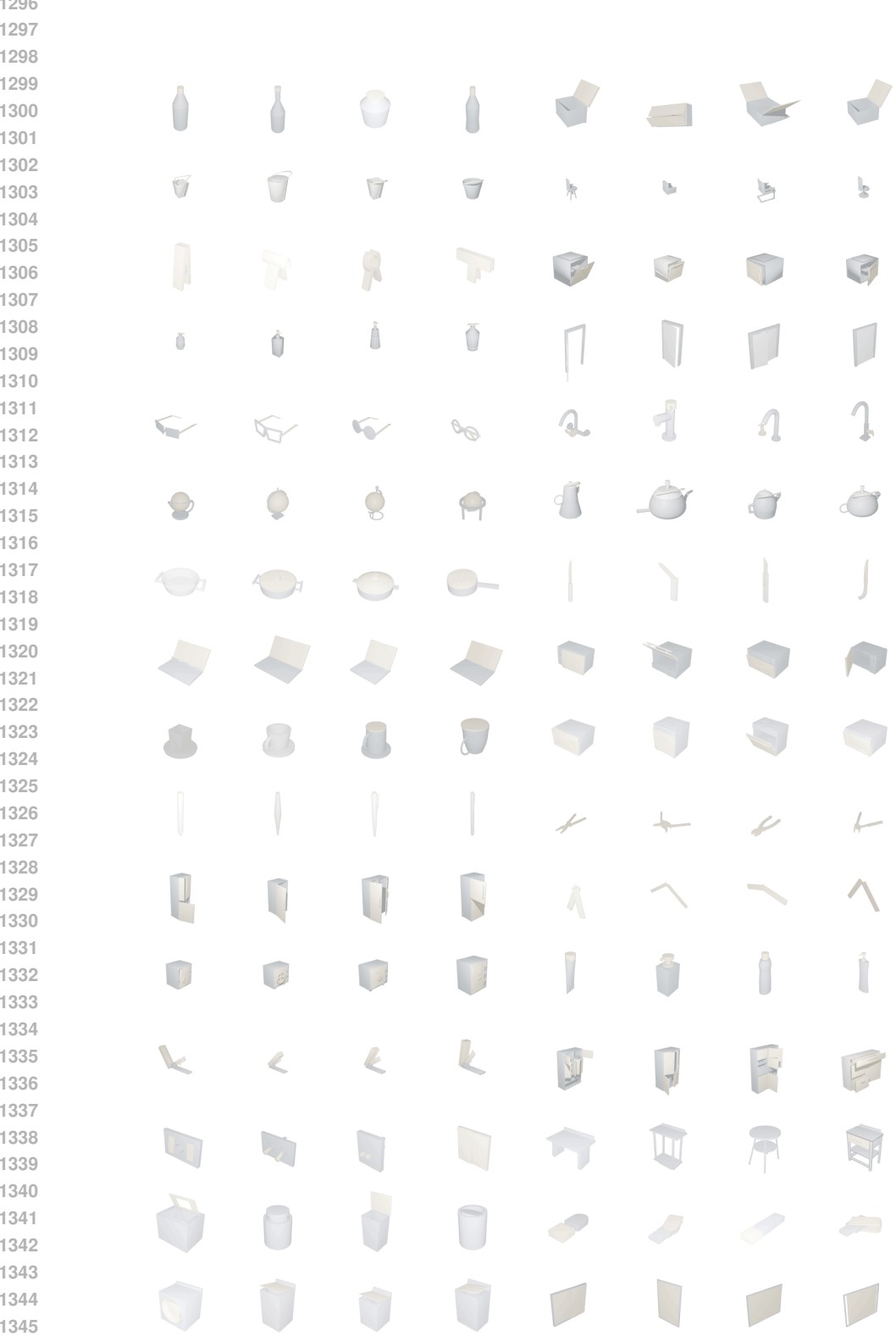

Figure 8: Visualization of synthesized objects.

