# OpenReview forum: "ArtiPG++: Towards Efficient Procedural Generation of Articulated Object Data"
_ICLR.cc/2026/Conference — ICLR 2026 Conference Withdrawn Submission_

### Official Review · Reviewer_Y8Ni · 2025-10-27

**Soundness:** 3
**Presentation:** 3
**Contribution:** 2
**Rating:** 4
**Confidence:** 2

**Summary:**

This paper introduces ArtiPG++, a fully procedural pipeline for generating large-scale datasets of realistic articulated objects with rich, precise annotations. Extending previous methods ArtiPG, ArtiPG++ can generate diverse assets and geometries without the need of external assets.

**Strengths:**

1. Efficient procedural generation of diverse and maintainable object structures.
2. It can generate realistic geometry details by generative models.
3. Enabling 10× faster generation in terms of human efforts needed.
4. The experiment is comprehensive.

**Weaknesses:**

1. The generative model for detailed geometries is also trained from real-world data with annotations. How is this dataset collected? Also, can the resulting model go beyond the dataset?
2. It is claimed that the method can synthesizes articulated object structures entirely from scratch. However, in the method section, a pre-defined Object Part Topology have to serve as input to the method?
3. Why ArtiPG baseline is not in Table 1? Also, can there be results of more categories in terms of synthesis quality?

**Questions:**

See weaknesses.

---

### Official Review · Reviewer_HAcB · 2025-10-31

**Soundness:** 2
**Presentation:** 3
**Contribution:** 2
**Rating:** 4
**Confidence:** 5

**Summary:**

The paper introduces Arti-PG++, a procedural method for articulated object generation. The authors manually define a template for each object (e.g, chair), the components (e.g., seat, back, leg), and connectivity between the parts, and the articulation parameters (i.e., the joints and joint parameters). Then, they sample the "spatial structure" of the objects via randomly sampling values like seat length and seat height to generate a rough object shape via primitive structures like cubes and cylinders. Then, a diffusion model takes the rough point cloud shape and generates a smooth object mesh.

**Strengths:**

- The paper studies an important problem.
- Procedural generation makes sense as we'd like to expand our dataset for pretraining robotic policies and other tasks.

**Weaknesses:**

- The spatial structures are randomized but notably the articulated parts (i.e., what are the joints and what are the joint parameters) are predefined via templates. To me, the joints are actually what make the "articulated" object generation uniquely interesting. There are many diffusion models that can generate diverse shapes and textures of objects so randomizing spatial values like seat length isn't interesting. The joint type and parameters are hard-coded, which is disappointing.



- The paper ignores many previous works including URDFormer, Real2code, Articulate-Anything, SINGAPO, GaussianArt to name a few.

- Also, there's some recent works such as PartCrafter that can simultaneously generate geometry and part segmentation instead of this two-stage approach (random sampling via template and then fill out geometric details via diffusion). The paper should compare against such methods.

- The introduction claims that "current methods for collecting articulated objects and their annotations are either based on human effort or physics simulators", which is false and confusing. There are many other automatic methods like NAP that the authors included, and many more that the authors ignored. The reference to "physics simulators" seems to be the works included in Sec 2.3. These works deal with "mobility perception" as defined in this great review paper: Survey on Modeling of Human-made Articulated Objects. They already assume available articulated assets in the simulator, and the primary goal is robot learning. I find it perplexing that the authors choose to include these methods while omitting so many recent works in automatic articulated object generation / reconstruction that could serve as baseline against his paper.


- The approach based on templates can be restrictive in some cases. For example, there isn't a single template for "faucet". Some faucets have one lever one while other have two. Not only the geometry but the functional movement is also ambiguous. For example, there are windows that open via a revolute joint while others open via a sliding joint. There isn't one template for the windows. The paper doesn't discuss this issue.

- Part 3.1. says that the template parameters are constrained by the values of their predecessor parts. This makes sense as for example, we don't want the chair seat to be much bigger or smaller than the back. But designing the constraints seems very non-trivial and object-dependent, requiring huge human labor. The paper also doesn't detail how to do this nor giving any examples of this process.


Nitpick:
Line 193: "psuedo" code.

**Questions:**

Please see weaknesses.

---

### Official Review · Reviewer_sJai · 2025-10-31

**Soundness:** 2
**Presentation:** 2
**Contribution:** 2
**Rating:** 6
**Confidence:** 4

**Summary:**

This paper studies the generation of articulated assets. First, a rule-based procedure generates the object structure in a parametric way. Then, the parametric model is sampled into a point cloud, and the full detailed geometry is generated based on the point cloud via a diffusion model. Comparison with prior articulated object generation models is presented and shows improved results.

**Strengths:**

- Generation of articulated assets is an interesting and important 3D vision problem for robotics.
- The method addresses this problem in an interesting procedural generation way.
- The detailed geometry is also generated with the kinematic structures.

**Weaknesses:**

- Currently, the structure is purely based on human-designed rules; no data-driven or learnable parameters are involved in such a procedure.
- Can this procedural generation be handled by modern VLMs?
- As far as I understand, the object geometry is generated as a whole — is there a way to get per-rigid-part shape?
- It’s not clear how such a model can be conditioned on images or partial parts/constraints, etc.

**Questions:**

Please refer to the weaknesses. The main concern is that the procedure is too limited and is not a data-driven or extensible approach, nor a VLM-based one. Please clarify or justify this.

---

### Official Review · Reviewer_PBBY · 2025-11-01

**Soundness:** 3
**Presentation:** 2
**Contribution:** 2
**Rating:** 4
**Confidence:** 3

**Summary:**

The paper proposes a pipeline for procedurally generating articulated objects. They first synthesize primitive geometric shapes that are part of the asset based on procedural rules, these shapes are then enhanced to have greater details using a 3D diffusion model.

**Strengths:**

- The authors motivate a few innovations over the ArtiPG pipeline to improve the object synthesis speed.
- The evaluation is thorough and the authors perform numerous downstream applicability analysis on visual affordance estimation and part segmentation.

**Weaknesses:**

- I believe the claim of not relying on external assets is inaccurate given that the diffusion model used to provide finer geometric details in the proposed approach is trained on a dataset of such assets? Do the authors have evidence of non obvious generalization over just the technique used in Arti-PG where a reference asset is sampled and geometric details from it are incorporated?
- The contribution (1) of the novel framework for procedural generation (L067) is not clearly substantiated in the paper. What are the key innovations here over ArtiPG – can you provide clear metrics to evaluate the proposed changes?
  - (L291-295) The user study suggests a new interface for the ArtiPG rather than a new algorithm? While this is still valuable and encouraging to see reduction in human effort spent, it will be useful if the authors clarify the exact differences/innovations compared to prior work in this aspect of the pipeline.

**Questions:**

- (Table 1) could you add how ArtiPG compares on the synthesis quality here? This will help address some concerns raised in the weaknesses.
- (Table 3) what exactly are performance numbers referring to – are those success rates?

---

### Note · Authors · 2025-11-13

I have read and agree with the venue's withdrawal policy on behalf of myself and my co-authors.